# The most at-risk regions in the world for high-impact heatwaves

**Vikki Thompson** [1]✉, **Dann Mitchell** [1], **Gabriele C. Hegerl** [2], **Matthew Collins** [3], **Nicholas J. Leach**[4,5] **& Julia M. Slingo** [1]

Heatwaves are becoming more frequent under climate change and can lead to thousands of excess deaths. Adaptation to extreme weather events often occurs in response to an event, with communities learning fast following unexpectedly impactful events. Using extreme value statistics, here we show where regional temperature records are statistically likely to be exceeded, and therefore communities might be more at-risk. In 31% of regions examined, the observed daily maximum temperature record is exceptional. Climate models suggest that similar behaviour can occur in any region. In some regions, such as Afghanistan and parts of Central America, this is a particular problem - not only have they the potential for far more extreme heatwaves than experienced, but their population is growing and increasingly exposed because of limited healthcare and energy resources. We urge policy makers in vulnerable regions to consider if heat action plans are sufficient for what might come.

Record-breaking temperature extremes can cause severe impacts on society and the environment, as was seen in western North America in June 2021[1–3]. Identifying which regions globally have perhaps been lucky not to have experienced higher temperature extremes so far is important and is the focus of this study. Often, regions are only prepared for events as extreme as they have already experienced, with planning initiated by past disasters. Policymakers and governments need to prepare for events beyond current records – particularly with trends caused by anthropogenic climate change enhancing the probability of extremes[4].

Heatwaves are deadly–but better preparation can save lives[5,6]. Planning ahead can reduce mortality from climatic extremes. For example, city heat plans that include actions such as establishing cooling centres or reducing hours of work for outdoor workers can reduce heat impacts. Policy changes following the 2003 European heatwave led to fewer deaths after the similar magnitude 2006 event[7], and humanitarian response plans in Bangladesh reduced mortality from Cyclone Amphan in 2020[8].

Understanding the likelihood of such extreme heat events is essential to allow society to prepare for them, but by their very definition, these events are rare[9]. The chance of extremes can be assessed using the observational record but, with global records of daily data only spanning the last century, it is hard to estimate reliable return periods for rare events[10,11].

When investigating extreme heat events, decisions must be made about how the extreme is measured[12]. We use the annual maximum value of daily maximum temperature (*TXx*), which is recommended by the World Meteorological Organisation (WMO) for assessing heatwaves[13]. There are many alternative climatic extreme measures, such as a count of (multiple) days above a threshold[14] or above a percentile[15]. Some regional studies use heat comfort indices, which combine temperature with humidity[16]. The minimum temperature may also be used–high nighttime temperatures prevent the body from cooling, increasing health impacts[17]. The alternative measures are often best suited to particular regions; as we carry out a global study we use *TXx*.

The aim of this study is to identify which regions globally have perhaps been lucky not to have experienced higher temperature extremes so far. We argue that these regions may be particularly vulnerable to the impacts of a record heatwave because there has been no need for adaptation thus far. We use extreme value theory (EVT) to assess return periods of observed temperature extremes globally. We

¹School of Geographical Sciences, University of Bristol, Bristol, UK. ²School of Geosciences, University of Edinburgh, Edinburgh, UK. ³Department of Mathematics and Statistics, University of Exeter, Exeter, UK. ⁴Atmospheric, Oceanic and Planetary Physics, Department of Physics, University of Oxford, Oxford, UK. ⁵Climate X, 1st Floor, 21 Great Winchester Street, London, UK. ✉e-mail: vikki.thompson@knmi.nl

begin by investigating the western North American heatwaves of June 2021 as an example of the technique in a region which has been shown to have experienced an event beyond the statistical maximum. We then use the same methods to assess daily heat extremes globally, identifying where in the world the current record has a short return period. We also identify regions where the observed record temperatures appear statistically implausible prior to their occurrence. Furthermore, we use results from the analysis of large ensembles of climate models to support the conclusions from the observational record.

EVT provides a statistical method to estimate the return periods of rare events−under the assumption that all events are from the same distribution[18]. In climate science, this technique is used to assess meteorological extremes in both observations and climate models[19–21]. When the event statistics are nonstationary, for example, because of external forcing such as greenhouse gas forcing, EVT can be extended to allow for this. When estimating the return period of climate extremes, to allow for climate change, a linear relationship with global mean surface temperature is often assumed. For example, when investigating the Siberian summer heatwave of 2020[22] and the southern USA flooding in August 2016[23]. This assumption may be invalid; the relationship can be non-linear and vary regionally. For example, non-linear interactions between soil moisture and surface temperature may affect the events[24]. Local forcings, such as aerosols or irrigation, may influence specific regions leading to inaccuracies when applying one method globally[25,26]. EVT assumes data points are independent; if local decadal variability prevents this, it may affect the results[27]. The historical records may not sample the full range of situations that give rise to extremes. In these cases, extrapolation to the rarest events without any additional knowledge of the physical mechanisms involved in such

extremes may lead to inaccuracies. Despite these limitations, EVT is considered the best practice for estimating extremes[11,27].

## Results

### Western North America heatwave, June 2021

In June 2021, western North America experienced a record-breaking heatwave. In Lytton, British Columbia, temperatures of 49.6 °C were observed on June 29th, breaking the previous record by almost 5 °C[28]. The heatwave was associated with an unusual circulation pattern, with a blocking anticyclone leading to a stagnant warm air mass[29].

A rapid attribution study found the event was so far beyond what had been previously observed that it was deemed virtually impossible without climate change[30]. In that study, the region assessed was chosen based on the record-breaking event itself−so by definition will appear particularly rare. In this study, we will make a global assessment of the risk of unprecedented heat. Therefore, we use a predetermined set of regions[31] (see Methods). As in[30], we use ERA5, a reanalysis dataset, as a proxy for observations[10] (see Methods). Data from 1959 to 2021 is assessed. For the June 2021 event, we use the region of Alberta, Canada, as that region is shown to have the largest extreme for the June 2021 event in terms of standard deviations from the mean[2]. As shown in Fig. 1, we find *TXx* for 2021 (which occurred on June 29th) is beyond the plausible range given by the EVT distribution−it would have been deemed extremely unlikely prior to its occurrence, in agreement with[30]. It should be noted that the distribution is shifted with a change in global mean surface temperature (GMST) rather than using GMST as a covariate.

We can investigate whether, prior to June 2021, the region appeared particularly susceptible to such a record-breaking event (Fig. 1c/d). The previous record event from 2018−now the second

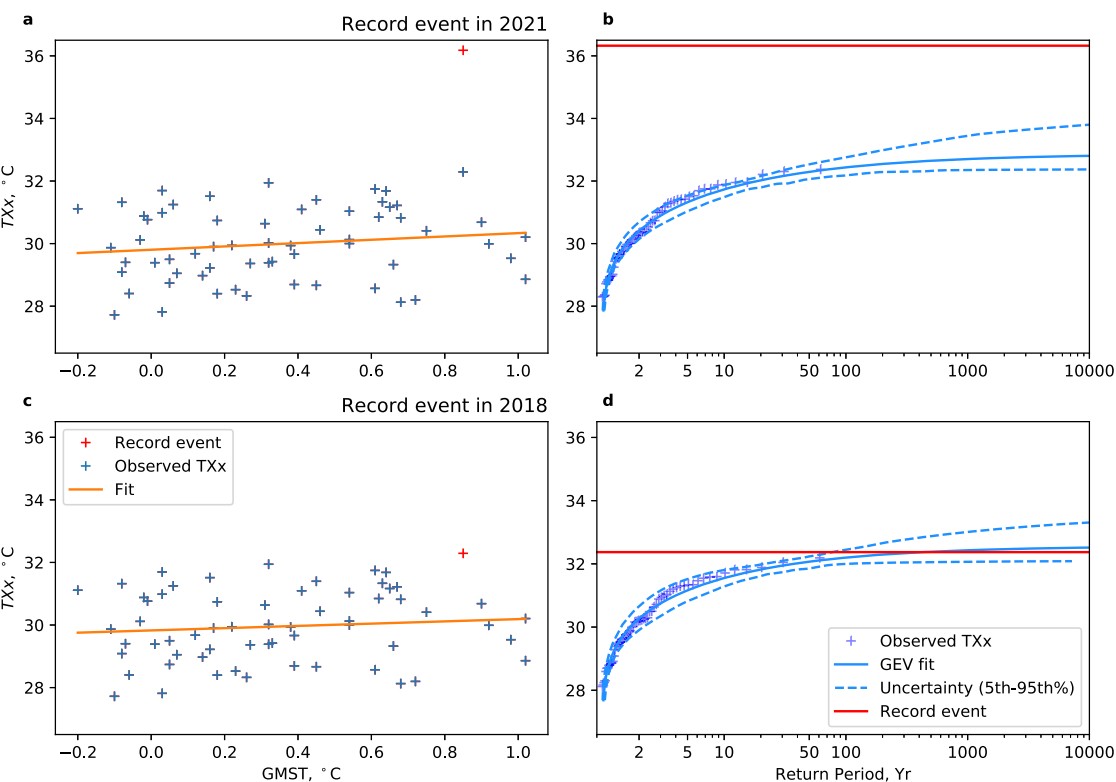

**Fig. 1 | Generalised Extreme Value (GEV) fit of an annual maximum of observed daily maximum temperature (*TXx*) for Alberta, Canada.** Data from ERA5[10], 1959–2021. **a** The observed *TXx* plotted against global mean surface temperature (GMST). The record event of 2021, shown in red, is not included in the fit line. **b** Return period plots adjusted to the current climate based on GMST = 1, calculated excluding the 2021 event, the solid blue line is the GEV fit, and dashed lines indicate

the 5th to 95th percentile uncertainty range. The red line indicates the 2021 record temperature. **c** as in (**a**), but with 2021 removed from observations, and the previous record, from 2018, shown in red and excluded from the fit line. **d** as in (**b**) but with a red horizontal line indicating the 2018 record temperature. Source data are provided as a Source Data file.

hottest *TXx* in the reanalysis period after adjustment by global mean surface temperature–is shown to have a 166-year return period. As may be expected, this is much closer to the length of the observations and is not a difficult record to exceed within this century, and society and ecosystems have not been exposed to large extremes recently. When applying an EVT fit to assess the return period of a specific event, it is normal practice that the event in question is excluded[27]. Usually, that event is the final event chronologically, as the assessment is likely triggered by the event itself[11]. Therefore, data from after the event cannot be used as it is yet to happen. We use a slightly different method, excluding only the record year (in this case, 2018) from the data from 1959 to 2020. This allows us to include more information about the distribution, thus giving a more accurate measure of the return period. We note that this does add selection bias to the analysis by excluding the record of records, but it helps to assess the sensitivity

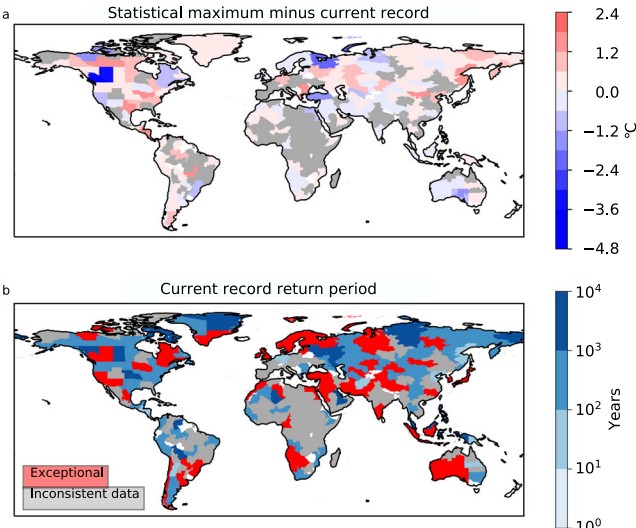

**Fig. 2 | Generalised Extreme Value (GEV) fit to calculate the return period of record events globally. a** The temperature difference between the statistical maximum event calculated by a GEV fit excluding the record and the current record. Grey regions indicate a lack of consistency between reanalysis datasets (see Methods). **b** The return period of the current record, when calculated by a GEV fit excluding the record. Red regions indicate where it is not possible to calculate a return period because the event falls outside of the GEV fit. Regions from Stone (2019)[31]. Source data are provided as a Source Data file.

of GEV fits to not sampling particularly strong extremes. This is important as other feedback may enhance the larger extremes, thus underestimating the tail shape when samples do not cover large events.

## Global assessment of reanalysis data

We can explore whether surprisingly extreme real-world events beyond the statistically plausible maximum occur in other regions globally or are unique to the western North America heatwave. This allows us to investigate the limitations of using EVT to assess return periods of extreme events and identify regions with low current records in terms of the return period.

We assess uncertainties in the reanalysis data by comparing two datasets–ERA5 and JRA55–and use only the regions where the extremes are consistent between the two datasets in the satellite era (1990 onwards), see Methods for further details[10,32]. The ERA5 data of daily maximum temperature, 1959–2021, are used to calculate a GEV fit of *TXx*, with the record event excluded from the calculation. The parameters of the GEV fit globally are shown in Fig. S1. Data from the years after the record event are included, as this provides more information about the true distribution.

For each region, we use the GEV fit to calculate a proxy for the statistical maximum–which we define as the magnitude of a 1 in the 10,000-year event–and compare this to the 1959–2021 record for that region (Fig. 2). Where possible we calculate the return period of the record event – though in regions where the record event is statistically implausible according to the GEV fit, as shown in Fig. 1b, no return period can be calculated with this method. Those regions where the current record events have lower return periods are at greater risk of the new records as high extremes have not been well sampled and may be at risk of experiencing events far beyond the current records.

The 2021 western North America heatwave is exceptional, almost 2 °C beyond any other region (Figs. 2a, 3b). But it is not the only region displaying a record event beyond the statistical fit; we find the current record is exceptional in 41 of the 136 regions, ~31% of the land surface (Fig. 2b). These implausible regions are spread across continents and latitudes–there is no apparent spatial discrimination. The events are also spread across the assessed time period but with more events in the later decades, possibly caused by greater availability of satellite data but potentially also by a non-linear signature of climate change (Fig. 3).

There are some regions which have not experienced events beyond 1-in-100-year events within the 62-year record (Figs. 2b, 3a, and

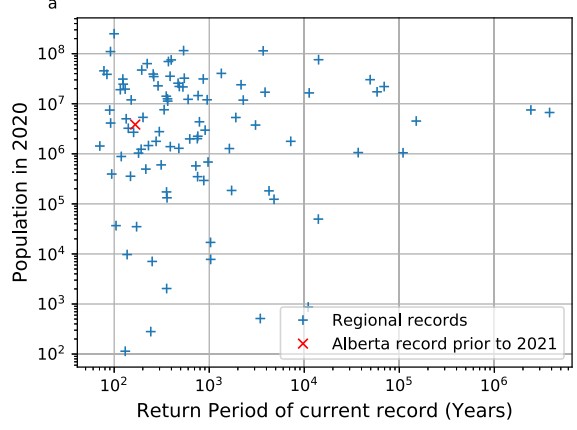

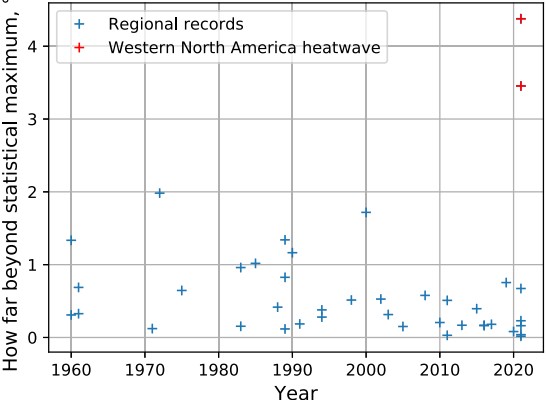

**Fig. 3 | Current record return period v population. a** Return period of the current record against the current population of the region[33] for the regions where a return period can be calculated. The red cross indicates the values for the Alberta region excluding the 2021 event (as in Fig. 1c, d). **b** Regions with statistically implausible

record events (where a return period cannot be calculated) plotted against the year of the event. Red crosses indicate two neighbouring regions of the 2021 Western North America heatwave. Source data are provided as a Source Data file.

**Table 1 | Regions where a record-breaking event is most likely**

| Region | Return period of record event (years) | Current record (°C) | 1-in-100 event (°C) | Record minus 1-in-100 (°C) | Population (2020, in millions) | Projected growth by 2050 (ssp5, %) |
|---|---|---|---|---|---|---|
| Russia, far eastern | 70.6 | 32.4 | 32.9 | **0.5** | 1.43 | 1.01 |
| **Central America**[a] | 78.1 | 36.2 | 36.6 | **0.4** | 45.47 | 1.05 |
| **Afghanistan** | 83.9 | 37.8 | 37.9 | 0.1 | 38.76 | **1.46** |
| **Papua New Guinea** | 89.6 | 32.5 | 32.6 | 0.1 | 7.49 | 1.26 |
| Central Europe[b] | 91.4 | 36.6 | 36.9 | 0.3 | **110.28** | 1.15 |
| Argentina, northwestern | 91.7 | 33.8 | 33.9 | 0.1 | 4.13 | 1.01 |
| Australia, Queensland | 94.2 | **44.2** | 44.3 | 0.1 | 0.40 | **1.66** |
| China, Beijing | 99.8 | 37.6 | 37.8 | 0.2 | **250.30** | 0.93 |

From the 136 regions where reanalyses datasets agree, the table lists the regions with a return period of a record below 100 years, that return period, the current record, adjusted to the present day using global mean surface temperature (GMST), 1-in-100 event magnitude, and population data[33]. See methods for further details.
[a]Region: Central American Integrated System, which includes Guatemala, El Salvador, Honduras, Nicaragua, Costa Rica, and Panama.
[b]Region: European Economic Area (central), which includes Germany, the Netherlands, Belgium, and Luxembourg.
Note: Regions in bold are developing countries (defined by the United Nations Human Development Index[34]), values in bold are the more noteworthy.

Table 1). These regions have had no need to adapt to such events and so may be more susceptible to the impacts of extreme heat. Statistically, these regions are also more likely to experience record-breaking extremes than other areas.

It is not only statistical likelihood which will affect the vulnerability of a region; this will also vary depending on socioeconomic factors. Here we use population[33] and economic development[34] projections as indicators to enable the potential risk to be qualitatively assessed (Fig. 3a). Regions with both a low return period and high population will have greater exposure to the hazard—the Beijing region of China has the highest population of all regions and is at risk statistically.

In Table 1, we list the regions which are statistically most at risk of a record heatwave. The table includes three developing regions, as defined by the UN Human Development Index (https://hdr.undp.org/). Afghanistan is the region of most concern as it is one of the least developed countries globally, with the historical record showing a low return period of ~80 years and steep projected population growth. The countries of the Central American Integration System region: Guatemala, El Salvador, Honduras, Nicaragua, Costa Rica, and Panama, are all developing countries. This region is vulnerable as, although the population is not expected to increase as much as elsewhere, the current record is further below the statistical maximum—suggesting the region could experience a large jump in the record. This is also the case for far eastern Russia (Khabarovsk region). Beijing, Hebei, and Tianjin provinces of China and Germany, Netherlands, and Belgium are vulnerable in terms of population number but, as developed countries, are more likely to have heat plans to mitigate potential impacts.

**Using large ensembles of climate models**

Climate models are a useful tool for assessing extreme meteorological events as, with multiple simulations possible, many times more data are available than for the real world, and so much rarer events that have been observed can be sampled[35,36]. The Coupled Model Intercomparison Project phase 6 (CMIP6) provides up to 50 ensemble members of several global climate models[37]. EVT methods have been used with climate models; for example, to compare return periods of daily extremes from the CMIP5 and CMIP6 historical model simulations, finding the two have no significant differences[38].

We use two 50-member climate model ensembles of CanESM5[39] and MIROC6[40] to investigate if the same behaviour of record event beyond the statistical fit is found. Since the models are not initialised from an observed state, they do not show the actual extreme events as have been observed. Instead, they model multiple realisations of the current climate and, therefore, the plausible pathways that the world's

weather could take, which may contribute to unprecedented extremes.

We assess the large ensembles using two different methods, described further in Methods. First, each ensemble member can be treated individually with a separate GEV fit, or all ensemble members can be merged into one single distribution (Fig. S2). It might be expected that when merging all ensemble members into one distribution, less statistically implausible extremes would be found—as the fit is better constrained with 50 times more information about the extremes.

In the first method, each model realisation has the same amount of information as is available for the real world. We find implausible extremes in at least 1 realisation for every region globally, with up to 50% of realisations showing the behaviour in some areas. Overall, we calculate GEV fits for over 10,000 distributions (50 realisations × 217 regions) and find exceptional—but meteorologically plausible—extremes in 26% of the regions for CanESM5 and 24% for MIROC6.

When merging all realisations to calculate just one GEV fit for each region, we find a similar proportion of exceptional extremes; 18 % of regions for CanESM5 and 22% for MIROC6 (Fig. 4b). The regions with implausible fits are different from the reanalysis, and only slightly fewer regions despite a much larger event set (Fig. 2b). As with observations, they are spread spatially across the world. This suggests that any region is susceptible to experiencing an extreme beyond the statistically fit—and it is not simply an artefact of the length of the observational record.

**Discussion**

Regions which have, so far, not experienced a particularly extreme event may be less prepared for the consequences of such an event. We have identified the regions where the current records have the lowest return periods (Table 1). In these regions, a record-breaking event is not only more likely but also likely to have greater impacts due to a lack of preparedness. Countries tend to prepare to the level of the greatest event they have experienced within collective memory.

Our global assessment of reanalysis data shows that statistically implausible extremes have occurred in 31% of regions between 1959 and 2021, with no apparent spatial or temporal pattern. It appears that such extremes could occur anywhere and at any time. When using climate model data to investigate further, we find 18–26% of regions in the model have the same characteristics. This suggests that everywhere needs to be prepared for a heatwave so extreme it is deemed implausible based on the current observational record. The June 2021 heatwave in western North America is shown to be exceptional in terms of how far beyond the expected it was. Although we highlight

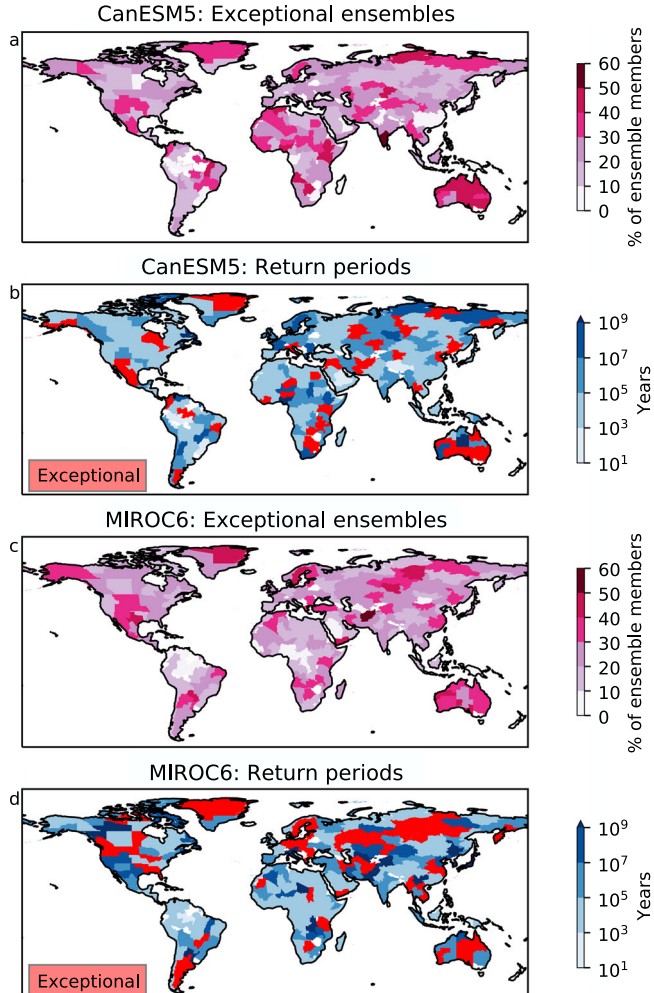

**Fig. 4 | Global map of records outside of Generalised Extreme Value (GEV) fit in climate model data. a** For a 50-member ensemble of CanESM5, 1959–2015, calculated using each ensemble member separately how many members show record events outside the GEV fit. **b** Taking all ensemble members to fit a single GEV distribution, the return period of the current record. Red regions indicate where it is not possible to calculate a return period because the event falls outside of the GEV fit. **c** As in (**a**), for a 50-member ensemble of MIROC6. **d** As in (**b**) for MIROC6. Regions from Stone (2019)[31]. Source data are provided as a Source Data file.

the processes inducing heatwaves and how these may change is vital to allow the risks to be quantified.

As we show, many heat extremes are beyond the statistical distribution in both observations and the model data; it is not possible to estimate the most extreme plausible events using EVT. Further research into the greatest plausible climatic extremes is essential to allow policymakers to plan for possible future events. Alternative methods for estimating the most extreme plausible events include assessing large ensembles of climate models[41] or finding regions which can be used as proxies for a specific location to increase the distribution[1]. Ensemble boosting methods, reinitialising climate model simulations of extreme events to find even more extreme plausible events, can enhance understanding[42]. A better understanding of the atmospheric dynamics causing the greatest extremes may also aid knowledge of the physical maximum temperature in some regions.

We show that in many regions, there is a disagreement between reanalysis products when assessing the most extreme days. These regions of disagreement span all continents, though they are of particular note over Africa. Reanalysis products provide spatially complete datasets of climate indicators globally, but in regions with less observational data available, such as Africa, they will perform worse[43,44]. Differences between reanalysis products have been previously identified[45]. Differences in methodology in coastal regions may have some impact, but this does not explain all regions[43]. Further investigations into the differences in extremes in reanalyses, and the causes, would be valuable.

Different heatwave metrics are more suited to different impacts, but in this study only *TXx* is considered. In some regions, extreme 5-day mean temperature levels will have greater impacts than daily extremes[13]. Heatwave metrics which incorporate humidity are useful for human health impacts[16], whereas temperature-only metrics are useful for infrastructure impacts[46]. Some metrics are better suited to specific regions due to, for example, land coverage. When assessing for adaptation purposes, a more thorough investigation would ensure findings are presented for policy-relevant timescales.

In conclusion, we have identified regions where record-breaking heat extremes are statistically more likely to occur because the current records do not sample the larger extremes well. Furthermore, these regions may be more susceptible to the impacts of such extremes due to a lack of preparedness. Based on both observational data and model data, we find that temperature extremes that appear statistically implausible based on the current observational record could occur in any region globally.

## Methods
### Data
We use reanalysis data and historical climate model simulations. The reanalysis datasets provide spatially complete gridded climate data by combining observational records with data from forecasting models and data assimilation systems used to fill gaps where direct observations are unavailable. We use ERA5 and JRA55 within our analysis[10,32]. The use of two different reanalysis datasets increases confidence in our results. Data are available from 1959 to the present (ERA5) and 1958 to the present (JRA55). We use the heat index of the highest daily maximum temperature of the year (*TXx*), recommended by the World Meteorological Organisation (WMO)[13]. To identify consistent regions, we compare the data from 1990 to 2022 only. For a region to be classed as consistent and included in our results, the record year from ERA5 must be in the top 5 years for JRA55. We use the global mean surface temperature (GMST) from the National Aeronautics and Space Administration (NASA) Goddard Institute for Space Science (GISS) surface temperature analysis[47,48].

Two large ensembles of global climate models, CanESM5 and MIROC6, are used[39,40], each having 50 realisations. We use historical simulations, 1950–2014. CanESM5 is comprised of an atmospheric

regions where a record event is statistically more likely (Table 1), we are not suggesting that these regions will experience events as extreme as the June 2021 heatwave.

The vulnerability of a region to the impacts of heat is not only dependent on the statistical likelihood of a record-breaking event. Socio-economic factors will make a large difference to the preparedness, with developing countries less likely to have adequate heat plans in place. Countries with greater projected population growth may be able to cope with current conditions but may find their health services and energy supply overwhelmed if policymakers do not plan adequately. We have highlighted some regions that may be most susceptible—with both a high statistical chance of a record and a rapidly increasing population.

Although changing dynamics could be a factor in the more recent events, we have shown extremes beyond the statistical fit, exceptional extreme events, occur throughout the reanalysis time period—so this does not fully explain the outliers. Further investigation into whether the underlying statistics of the distribution are shifting, resulting in past observations no longer being useful for assessing future, or even current, risk of extremes, is needed. Increasing our understanding of

general circulation model with ~2.8° resolution, an ocean general circulation model with ~1° resolution, a land surface scheme, explicit land and ocean carbon cycle models, and a sea ice model[39]. MIROC6 is comprised of an atmospheric model with a ~1.4° resolution, with land, ocean, and sea ice model components[40]. Compared to other CMIP6 models, both CanESM5 and MIROC6 have a coarse resolution, this allows a large ensemble to be produced more readily. It is often shown that as the resolution of models is increased, their performance improves[49]. For daily maximum temperature, this has not been shown —the CMIP6 ensemble shows no improvement compared to CMIP5, despite the increased resolution[20].

### Regions

We must take care when the spatial scale assessed is appropriate—for climatic extremes, results can differ vastly between spatial scales. Information about localised extremes can be found in individual station datasets, but this information is not available globally, and stations differ in quality and temporal coverage. Global-scale Earth System Models simulate climate over relatively large grid boxes, which must be considered if comparing observational data and model data. Increasing the spatial scale will dampen the absolute magnitude of an extreme—a balance must be reached. A study defined five sets of regions[31] designed for assessing climatic extremes at different spatial scales from 0.1 to 10 Mm². The regions are based on societal impacts, with political and economic boundaries used. This enables results to be aligned with policymaking. Similar sets of regions have been used to assess daily extremes already, for example, daily extremes in reanalysis datasets using 2 Mm² scale regions[50] and 0.5 Mm² in our earlier work[2]. Using regions of 2 Mm² or smaller ensures extremes caused by mid-latitude synoptic-scale weather systems will be included.

We use the predefined 0.5 Mm² regions; this scale corresponds to an area with a diameter of ~800 km[31]. There are 237 regions in the dataset, but because of the way, the regions are defined, using political boundaries based on impacts, some areas of the world are excluded. The main areas missed are Armenia, the Balkans, Bangladesh, most Caribbean islands, Belarus, Georgia, Nepal, New Zealand and most other Pacific Islands, North Korea, and Sri Lanka. We remove the Antarctic regions, leaving 217 regions. This is further reduced to 136 regions when excluding regions where the two reanalyses products are not consistent.

### Extreme value theory

We use extreme value theory (EVT) to assess return periods. We use block maxima—taking the highest daily maximum temperature of each year. Block maximum is better for temperature as a peak over the threshold is heavily impacted by clustering[27]. When a heatwave spans several days, it will get double-counted by a peak over the threshold. Block maxima also take away any subjective choice over the threshold limit to use.

We use the heat index of the highest daily maximum temperature of the year (TXx). It is assumed that the scale and shape parameters of the distribution are constant, as in other assessments of heat extremes[22,30]. To allow for changes in the location parameter through time due to warming globally, we calculate the linear relationship between the regional extremes and GMST and adjust the regional data to a GMST of 1 °C. This assumes global warming is the main factor affecting the extremes beyond natural variability and leads to results which are at approximately the current warming level (as in[11,27]).

We include all data except the record year in the calculation of the GEV distribution. Commonly the data after the event are disregarded— as it is often the most recent event that is being assessed, there will be no later data to use anyway. If we were to exclude data from after the record event, some regions where the record occurs early in the time series might have a GEV fit based on very few data points leading to large uncertainties. Including data from after the extreme makes the regional results more comparable—every GEV fit is calculated from the same number of years of data. Uncertainty in the GEV fit is included by bootstrapping; we randomly select data from the distribution 100 times, recalculating the GEV fit each time. The 5th to 95th percentile range is used to represent the uncertainty (as in Fig. 1b, d).

There has been discussion over whether the observed extreme event should be included in the statistical fit[27,30]. The record event has been observed and therefore adds knowledge to the distribution of extremes, and without its inclusion, the return periods of extreme events may be biased low. In most previous studies, the record event has not been included in assessments as by including it, there may be selection bias[19,20,22,23]. However, without the inclusion of the event in question, some recent events have been so extreme they are beyond the statistical maximum, defined in this paper as the 1-in-10,000-year event calculated from EVT without the event itself included. This is the case for the western North American heatwave of June 2021[30].

The 1-in-10,000-year return level is taken as the statistical maximum. This value is taken as an approximation of the asymptote value for the curve and is chosen as, by this point, the curve is approximately horizontal; increasing beyond makes little difference to the values. The choice of 10,000 years—rather than anything longer—makes little difference to the value.

We apply GEV to fit historical simulations of climate models. For each of the two models used, there are 50 ensemble members. We adjust each model by its own ensemble mean GMST, thus dampening internal variability and allowing for differences between the modelled and observed trends. When applying the GEV fit, we use two different techniques. The first takes each ensemble member individually to fit a GEV distribution, providing 50 different fits for each of the 217 regions globally. In the second technique, we use all 50 ensemble members to calculate one fit—taking TXx from every year of every ensemble member.

## Data availability

ERA5 data for surface temperature was downloaded from the European Centre for Medium-Range Weather Forecasts (ECMWF), Copernicus Climate Change Service (C3S) at Climate Data Store (CDS; https://cds.climate.copernicus.eu/). The Japanese 55-year Reanalysis (JRA-55-) data are available at the JRA project website (http://search.diasjp.net/en/dataset/JRA55). CanESM5 and MIROC6 data are available from the CMIP6 search interface (https://esgf-node.llnl.gov/search/cmip6/). Population data are available from the UN Human Development Report, 2022. (https://hdr.undp.org/). The shapefiles for the regions used to assess globally are available in the supplementary data of Stone, D.A.A. hierarchical collection of political/economic regions for analysis of climate extremes. Clim. Change 10 (2019) (https://link.springer.com/article/10.1007/s10584-019-02479-6). Source data are provided in this paper.

## Code availability

The code used to generate the figures in this paper and the Supplementary Materials is available from Github and Zenodo https://doi.org/10.5281/zenodo.7692244 and https://doi.org/10.5281/zenodo.6325508. All data needed to evaluate the conclusions in the paper are present in the paper and/or the Supplementary Materials.

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

## Acknowledgements

This work was supported by NERC project EMERGENCE (NE/S005242/1) (V.T., D.M., G.H., and M.C.) and NERC grant NE/L002612/1 (N.L.).

## Author contributions

V.T. and D.M. designed the study. V.T. performed the data analysis. V.T., D.M., G.H., M.C., N.L., and J.S. contributed to the interpretation of the result and the writing of the article.

## Competing interests

The authors declare no competing interests.
