## [Peer Review File · Nature Communications]

Reviewers' Comments:

Reviewer #1:

Remarks to the Author:

After an in-depth reading of the article, I consider it to be a relevant work of public interest. It is particularly interesting for its relevance to environmental risk management policies.

In relation to the extreme temperature risk maps, they point in the same direction as other research in the field of climate change.

In particular, the authors rightly point to the suggestion that areas that have not experienced such extreme events are particularly vulnerable in the absence of specific plans in place.

Overall, I consider this to be a rigorous piece of work and consider it suitable for publication.

Reviewer #2:

Remarks to the Author:

This is an interesting paper, and I love the concepts. The problem of regions "getting lucky" and not preparing for heatwaves is a critical one, and it is great to see this highlighted.

In terms of feedback, I think it is sufficient to accept the paper with minor revisions. The paper's conclusions seem to call for a more in-depth analysis of this problem, but those additional analyses are likely beyond the scope of this paper. I would therefore suggest that the authors make several general revisions to the introduction and discussion/conclusion in terms of framing.

First, it seems as though one of the most critical findings of the paper is that GEV fits are not good approximations of our risk of extreme heat events! If the authors find implausible extremes in at least one of the CMIP ensembles for every region globally, then perhaps the GEV definition of "implausible" and a 10,000 year return period is questionable. It would be good provide further defense in the introduction section about why GEV fits are considered best practice for estimating extremes, and the choices you made when fitting the model.

It would be good to include greater discussion of the GEV limitations in the discussion/conclusion section, including suggestions about how people can better understand what plausible extremes might look like in their region. What alternative techniques could help with this?

Secondly, you mention briefly in the conclusion that TXx is not a good proxy for heatwave impacts in many regions. It would be good to add further discussion of this in the intro and conclusion, explaining what kind of impacts are associated with TXx and which are not. It would also be good to comment on the large regions of the world for which the reanalyses do not agree on TXx. What does this mean for our understanding of heat extremes in these regions? Can we believe the CMIP results for those regions?

Smaller comments:

At first, I did not understand how a region in Figure 2 could be white in plot (a) but exceptional/red in plot (b). I eventually realized that some of the white regions are slightly below 0, and therefore those regions have observed extremes that are slightly above the 10,000 year threshold. Examples of these regions include Namibia, Western Australia, some of Saudi Arabia, etc. All this to say, you might consider adjusting the plot colors so that there is not a white category that encompasses both positive and negative values in plot (a), to make things a little easier for the reader.

Figure 3b – why is the vertical axis negative? Label is "how far beyond statistical maximum", so reader would assume that the axis would be positive showing degrees beyond the statistical maximum.

Table 1: What does the bold refer to?

Reviewer #3:

Remarks to the Author:

The study analyses observational and model data and uses Extreme Value Theory to identify regions of the globe, where very rare hot extremes are statistically more likely to occur, according to estimated return periods. A large part of the globe was excluded from observational analysis due to inconsistency between different reanalysis data bases, as regards annual daily maximum temperature (TXx). From the realization of large ensembles of climate models over 217 regions, it was found that temperature extremes beyond the statistical fit (actually statistically implausible based on the current record) can occur in any region globally. The extreme event of June 2021 in Northern America was first used to test the methodology.

Despite obvious uncertainties and assumptions in the analysis, I think it is a very interesting paper with potentially very high impact, calling for global awareness and preparedness against severe thermal risk, especially from unprecedented extreme events in vulnerable regions.

There are a few points and suggestions that might be helpful to the authors for the improvement of the manuscript.

I was a little a bit confused at the beginning of the reading about the real scope of this study. This should be better and more precisely conveyed at the beginning and throughout the text. For instance

Lines 40-42: 'lucky' regions. Here, the authors give the impression that they seek lucky regions of the globe and this is the focus of the study. According to the title of the manuscript (not sure that is very suitable), they identify most at risk regions (rather unlucky). I think that the focus and scope of the study should be described in a more accurate way and further highlighted throughout the text.

In general, the introduction seems somewhat 'patchy', including also methodological approaches, discussion etc. I think the Introduction should be rearranged, also with richer literature review.

The scope and focus of the study should be more clearly described at the end of Introduction, along with a very brief description of methodological steps, in order to prepare the reader.

The reader misses the accurate 'take home message' which should be also better conveyed in the manuscript

Lines 14-16: please rephrase

Line 22. add 'temperature' before 'record'

Lines 47-49: it can be omitted, just focus on EVT on climate study

Line 58: TXx is mentioned here but not yet introduced in the text.

Lines 57-58: Is it assumed by the authors or in literature generally? For the latter case provide reference. Obviously the relationship between Txx (the annual maximum temperature) and GSMT is not linear. Also, warming rates vary strongly across the globe.

Lines 60-70. The authors speculate in a way, not being confident about the correct procedure.

Maybe transferred in methods? Or discussion? The choice of methods should be better justified in general.

Lines 67-68: What do you mean by 'historically'? As previously reported in literature? Then provide more references.

Lines 91: define the observational period

Lines 104-105: Why is usually the last event in the record? Please explain

Line 131: 1959-2021 or 1959-2022?

Figure 1 should be further commented in the text. Is the red cross in c) the 2018 record? (not included in the fit though)..? Does the red horizontal line in d) represents 2018 Txx? If so, this should be included in the caption

Figure 3: the description of the caption could be improved. In 3b, can you indicate with a red cross the 2021 event? Also, only negative values of the difference between the statistical maximum and current record are included in Fig. 3b. Both positive and negative values appear in fig. 2a. Any comment on this?

EVT and GEV: maybe some additional information about the application and limitations/uncertainties would be useful. E.g. assumptions on data distribution, assumption about scale and shape parameters (constant values?), estimation methods (MLE or PWM) and possibly others.

263-264: why 5 and not 6 or 4? any reference?

Is the number of the selected regions 116 (line 294), 136 (line 139), 217 or 27 for models (line

327)?

Lines 309-310: Temperature records can occur anytime within the observational period, even in the very early years. Why the authors think this will be the very recent event?

The authors should pay some extra attention to the text. Although language is good, missing commas (,) make it difficult to understand in some parts of the text.

Reference list requires additional attention, too. Please use a homogenized format for references. It changes many times across the list (e.g. the order of initials and surnames, DOI, 'and' for the last author, the location of publication year and many more....). Also,

Ref No 35 is incomplete

References No 17 and 26 are the same.

In some references the authors use 'et al' (e.g. ref No 6 or 21), but in other references a long list of authors is included (e.g. ref No 30).

Philip et al. (2022) in the text but (2021) in Ref. list

Response to reviews: The most at-risk regions in the world for high-impact heatwaves

We thank the reviewers for assessing our manuscript and for their constructive feedback. We have made changes based on their comments – particularly restructuring the introduction and including a broader literature review. Below is a point-by-point response to the reviews, including the three reviews (in red), and our response (in black, quotes from the manuscript in italics). The updated manuscript has changes highlighted in red.

Reviewer #1 (Remarks to the Author):

After an in-depth reading of the article, I consider it to be a relevant work of public interest. It is particularly interesting for its relevance to environmental risk management policies.

In relation to the extreme temperature risk maps, they point in the same direction as other research in the field of climate change.

In particular, the authors rightly point to the suggestion that areas that have not experienced such extreme events are particularly vulnerable in the absence of specific plans in place.

Overall, I consider this to be a rigorous piece of work and consider it suitable for publication.

Thank-you

Reviewer #2 (Remarks to the Author):

This is an interesting paper, and I love the concepts. The problem of regions “getting lucky” and not preparing for heatwaves is a critical one, and it is great to see this highlighted.

In terms of feedback, I think it is sufficient to accept the paper with minor revisions. The paper’s conclusions seem to call for a more in-depth analysis of this problem, but those additional analyses are likely beyond the scope of this paper. I would therefore suggest that the authors make several general revisions to the introduction and discussion/conclusion in terms of framing.

First, it seems as though one of the most critical findings of the paper is that GEV fits are not good approximations of our risk of extreme heat events! If the authors find implausible extremes in at least one of the CMIP ensembles for every region globally, then perhaps the GEV definition of “implausible” and a 10,000 year return period is questionable. It would be good provide further defense in the introduction section about why GEV fits are considered best practice for estimating extremes, and the choices you made when fitting the model.

Further discussion of the limitations of GEV have been added to the introduction, along with references which support the use of GEV as a best practice. In the methods section the choices made when fitting the model are now discussed in more depth.

It would be good to include greater discussion of the GEV limitations in the discussion/conclusion section, including suggestions about how people can better understand what plausible extremes might look like in their region. What alternative techniques could help with this?

The following paragraph has been added to the discussion:

As we show many heat extremes are beyond the statistical distribution, in both observations and the model data, it is not possible to estimate the most extreme plausible events using GEV. Further research into the greatest plausible climatic extremes is essential to allow policy makers to plan for possible future events. Alternative methods for estimating the most extreme plausible events include assessing large ensembles of climate models (Kelder et al 2022), or finding regions which can be used as proxy for a

specific location to increase the distribution (McKinnon & Simpson, 2022). Ensemble boosting methods, reinitialising climate model simulations of extreme events to find even more extreme plausible events, can enhance understanding (Gessner et al 2021). Better understanding of the atmospheric dynamics causing the greatest extremes may also aid knowledge of a physical maximum temperature in some regions.

Secondly, you mention briefly in the conclusion that TXx is not a good proxy for heatwave impacts in many regions. It would be good to add further discussion of this in the intro and conclusion, explaining what kind of impacts are associated with TXx and which are not. It would also be good to comment on the large regions of the world for which the reanalyses do not agree on TXx. What does this mean for our understanding of heat extremes in these regions? Can we believe the CMIP results for those regions?

Text has been added to the introduction, explaining the choice of TXx as a measure of heat extremes in this study:

When investigating extreme heat events decisions must be made about how the extreme is measured (Perkins & Alexander, 2013). We use annual maximum value of daily maximum temperature (TXx), which is recommended by the World Meteorological Organisation (WMO) for assessing heatwaves (Klein Tank & Zwiers, 2009). There are many alternative climatic extreme measures, such as count of (multiple) days above a threshold (Sherwood & Huber, 2009), or above a percentile (Fischer & Schär, 2010). Some studies use heat comfort indices, which combine temperature with humidity (Di Napoli et al., 2021). Minimum temperature may also be used – high nighttime temperatures prevent the body from cooling increasing health impacts (Mukherjee & Mishra, 2018). The alternative measures are often best suited to particular regions, as we carry out a global study we use TXx.

The discussion of impacts in the discussion sections has been expanded on, and further comment of regions with disagreement between reanalyses has been added:

We show that in many regions there is disagreement between reanalysis products when assessing the most extreme days. These regions of disagreement span all continents, though are of particular note over Africa. Reanalysis products provide spatially complete datasets of climate indicators globally, but in regions with less observational data available, such as Africa, they will perform worse (Green, 2022; Gleixner et al., 2020). Differences between reanalysis products have been previously identified (e.g. Sheridan et al., 2020). Green (2022) suggest differences in methodology in coastal regions may have some impact, but this does not explain all regions. Further investigations into the differences in extremes in reanalyses, and the causes, would be valuable.

Different heatwave metrics are more suited to different impacts, but in this study only TXx is considered. In some regions extreme 5-day mean temperature levels will have greater impacts than daily extremes (Klein Tank & Zwiers, 2009). Heatwave metrics which incorporate humidity are useful for human health impacts (Di Napoli et al., 2021), whereas temperature only metrics are useful for infrastructure impacts (Matthews et al., 2022). Some metrics are better suited to specific regions due to, for example, land coverage. When assessing for adaptation purposes a more thorough investigation would ensure findings are presented for policy relevant timescales.

Smaller comments:

At first, I did not understand how a region in Figure 2 could be white in plot (a) but exceptional/red in

plot (b). I eventually realized that some of the white regions are slightly below 0, and therefore those regions have observed extremes that are slightly above the 10,000 year threshold. Examples of these regions include Namibia, Western Australia, some of Saudi Arabia, etc. All this to say, you might consider adjusting the plot colors so that there is not a white category that encompasses both positive and negative values in plot (a), to make things a little easier for the reader.

The colorbar has been altered so as not to include a white section. Now all regions which are exceptional in Fig.2b are a shade of blue in Fig.2a. Updated Fig.2 shown here:

Figure 3b – why is the vertical axis negative? Label is “how far beyond statistical maximum”, so reader would assume that the axis would be positive showing degrees beyond the statistical maximum.

Thank-you for this suggestion. Indeed, it would make better sense to switch the axis on Fig.3b. This has been updated as shown below:

Table 1: What does the bold refer to?

The countries in bold are developing countries, as defined by the UN Human Development Index. The values are the greatest ones. Text has been added to the caption:

Regions in bold are developing countries (defined by the UN Human Development Index, <https://hdr.undp.org/>), values in bold are the more noteworthy.

Reviewer #3 (Remarks to the Author):

The study analyses observational and model data and uses Extreme Value Theory to identify regions of the globe, where very rare hot extremes are statistically more likely to occur, according to estimated return periods. A large part of the globe was excluded from observational analysis due to inconsistency between different reanalysis data bases, as regards annual daily maximum temperature (TXx). From the realization of large ensembles of climate models over 217 regions, it was found that temperature extremes beyond the statistical fit (actually statistically implausible based on the current record) can occur in any region globally. The extreme event of June 2021 in Northern America was first used to test the methodology.

Despite obvious uncertainties and assumptions in the analysis, I think it is a very interesting paper with potentially very high impact, calling for global awareness and preparedness against severe thermal risk, especially from unprecedented extreme events in vulnerable regions.

There are a few points and suggestions that might be helpful to the authors for the improvement of the manuscript.

I was a little a bit confused at the beginning of the reading about the real scope of this study. This should be better and more precisely conveyed at the beginning and throughout the text. For instance Lines 40-42: 'lucky' regions. Here, the authors give the impression that they seek lucky regions of the globe and this is the focus of the study. According to the title of the manuscript (not sure that is very suitable), they identify most at risk regions (rather unlucky). I think that the focus and scope of the study should be described in a more accurate way and further highlighted throughout the text.

The abstract has been changed to include the term 'at-risk':

We use extreme value statistics to examine where regional temperatures records around the world might not have experienced their most extreme levels, and therefore communities might be more at-risk.

The beginning introduction has been restructured to better highlight the aim of the study. The most at risk regions that we highlight are the regions which have, so far, been lucky. They are not (necessarily) regions at risk of high *magnitude* heatwaves, but those regions which are statistically more likely to experience a high *impact* heatwave, due to lack of preparedness based on past experience of that region.

In general, the introduction seems somewhat 'patchy', including also methodological approaches, discussion etc. I think the Introduction should be rearranged, also with richer literature review.

The introduction has been restructured, to better highlight the aims of the study and expand the literature review. Changes are highlighted in the manuscript. Parts of the introduction have been moved to Methods.

The scope and focus of the study should be more clearly described at the end of Introduction, along with a very brief description of methodological steps, in order to prepare the reader.

The final paragraph of the introduction has been rewritten:

The aim of this study is to identify which regions globally have perhaps been lucky not to have experienced higher temperature extremes so far. We argue that these regions may be particularly vulnerable to the impacts of a record heatwave because there has been no need for adaptation thus far. We use EVT to assess return periods of observed temperature extremes globally. We begin by investigating the western North America heatwaves of June 2021 as an example of the technique in a region which has been shown to have experienced an event beyond the statistical maximum. We then use the same methods to assess daily heat extremes globally, identifying where in the world the current record has a short return period. We also identify regions where the observed record temperatures appear statistically implausible prior to their occurrence. Furthermore, we use results from analysis of large ensembles of climate models to support the conclusions from the observational record.

The reader misses the accurate 'take home message' which should be also better conveyed in the manuscript

Lines 14-16: please rephrase

Changed to:

In 31 % of regions examined, the observed daily maximum temperature record is exceptional. Climate models suggest that similar behaviour can occur in any region.

Line 22. add 'temperature' before 'record'

Added

Lines 47-49: it can be omitted, just focus on EVT on climate study

Sentence removed, and 'this technique' added to following sentence.

Line 58: TXx is mentioned here but not yet introduced in the text.

Full description has been added to the first use of the acronym - annual maximum value of daily maximum temperature (TXx).

Lines 57-58: Is it assumed by the authors or in literature generally? For the latter case provide reference. Obviously the relationship between Txx (the annual maximum temperature) and GSMT is not linear. Also, warming rates vary strongly across the globe.

This paragraph has been rewritten with more detail of the limitations of the method, and references to support method decisions:

As the climate has changed over the observational record a linear relationship between TXx and the global mean surface temperature is assumed. Some studies have shown this assumption may be invalid, and the relationship can be non-linear and vary regionally (Chan et al., 2019). Non-linear interactions, for example between soil moisture and surface temperature, may affect the events. Local forcings, such as aerosols or irrigation, may influence specific regions leading to inaccuracies when applying one method globally (Wild, 2009; Puma and Cook, 2010). EVT assumes data points are independent, if local decadal variability prevents this it may affect the results (Philip et al., 2020). The historical records may not sample the full range of situations that give rise to extremes. In these cases, extrapolation to the rarest events, without any additional knowledge of the physical mechanisms involved in such extremes, may lead to inaccuracies.

Despite these limitations, GEV is considered best practice for estimating extremes (Philip et al., 2020).

Lines 60-70. The authors speculate in a way, not being confident about the correct procedure. Maybe transferred in methods? Or discussion? The choice of methods should be better justified in general.

This paragraph has been moved to the methods, and the methods section has been expanded to better justify methodological choices.

Lines 67-68: What do you mean by 'historically'? As previously reported in literature? Then provide more references.

As in previous literature, this has been reworded and further references have been added to support the statement:

In most previous studies the record event has not been included in assessments as by including it there may be selection bias (Emanuel, 2017; Wehner et al., 2020; Ciavarella et al., 2021, Van der Wiel et al., 2017).

Lines 91: define the observational period

Added: *Data from 1959 to 2021 is assessed.*

Lines 104-105: Why is usually the last event in the record? Please explain

When investigating a specific event, it is generally soon after the event has occurred, to estimate the return period and understand how 'unlucky' a region has been. An explanation and reference has been added to the text:

Usually that event is the final event chronologically as the assessment is likely triggered by the event itself (van Oldenborgh et al., 2021). Therefore, data from after the event cannot be used as it is yet to happen.

Line 131: 1959-2021 or 1959-2022?

2021 – updated accordingly.

Figure 1 should be further commented in the text. Is the red cross in c) the 2018 record? (not included in the fit though)..? Does the red horizontal line in d) represents 2018 Txx? If so, this should be included in the caption

The figure caption updated to better explain Fig.1c) and d), and these figures are now referred to directly in the text.

Title of Fig.1c showed incorrect year, this has been corrected.

Figure 3: the description of the caption could be improved. In 3b, can you indicate with a red cross the 2021 event? Also, only negative values of the difference between the statistical maximum and current record are included in Fig. 3b. Both positive and negative values appear in fig. 2a. Any comment on this?

Following a comment from another reviewer Fig.3b has been altered to show x-axis data as positive values (how far above the statistical maximum, rather than statistical maximum minus current record, as it was in original submission).

Fig.3b includes only the data points where the return period cannot be calculated – therefore these points must all have a record beyond the statistical maximum (show as negative in Fig.2a). The figure and caption have been updated:

Figure 3: Current record return period v population. a) Return period of current record against the current population of the region, for the regions where a return period can be calculated. Red cross indicates the values for the Alberta region excluding the 2021 event (as in Fig.1c/d) **b)** Regions with statistically implausible record events (where a return period cannot be calculated) plotted against year of event. Red crosses indicate to neighbouring regions of the 2021 Western North America heatwave.

EVT and GEV: maybe some additional information about the application and limitations/uncertainties would be useful. E.g. assumptions on data distribution, assumption about scale and shape parameters (constant values?), estimation methods (MLE or PWM) and possibly others.

The paragraph on limitations of the EVT method in the introduction has been expanded:

When estimating the return period of climate extremes, to allow for climate change a linear relationship with global mean surface temperature is often assumed. For example, when investigating the Siberian summer heatwave of 2020 (Ciavarella et al., 2021) and southern USA flooding in August 2016 (Van der Wiel et al., 2017). Some studies have shown this assumption may be invalid, and the relationship can be non-linear and vary regionally (Chan et al., 2019). Non-linear interactions, for example between soil moisture and surface temperature, may affect the events. Local forcings, such as aerosols or irrigation, may influence specific regions leading to inaccuracies when applying one method globally (Wild, 2009; Puma and Cook, 2010). EVT assumes data points are independent, if local decadal variability prevents this it may affect the results (Philip et al., 2020). The historical records may not sample the full range of situations that give rise to extremes. In these cases, extrapolation to the rarest events, without any additional knowledge of the physical mechanisms involved in such extremes, may lead to inaccuracies. Despite these limitations, EVT is considered best practice for estimating extremes (Philip et al., 2020, van Oldenborgh et al., 2021).

A paragraph about alternative approaches to investigate plausible extremes has been added to the discussion. The methods section has been expanded to discuss estimation methods and assumptions.

263-264: why 5 and not 6 or 4? any reference?

Initially we evaluated where the record events were identical in both reanalyses, but very few regions were consistent across both reanalyses datasets. Expanding to 5 ensured we did not remove regions

where the top few extremes are very similar in magnitude and so may differ in ranking between the two reanalyses. No reference for this.

Is the number of the selected regions 116 (line 294), 136 (line 139), 217 or 27 for models (line 327)?

Corrected in line 294, it is 136 for reanalysis.

Corrected in line 329, 217 regions for model.

The models include more regions, as for the reanalysis it is only regions consistent between the two different datasets that are included.

Lines 309-310: Temperature records can occur anytime within the observational period, even in the very early years. Why the authors think this will be the very recent event?

In lines 309-310 we say 'as it is often the most recent event that is being assessed there will be no later data to use anyway', referring to the fact that events are often studied soon after they have occurred, rather than only events in the last few years.

The authors should pay some extra attention to the text. Although language is good, missing commas (,) make it difficult to understand in some parts of the text.

Reference list requires additional attention, too. Please use a homogenized format for references. It changes many times across the list (e.g. the order of initials and surnames, DOI, 'and' for the last author, the location of publication year and many more....). Also,

Ref No 35 is incomplete

References No 17 and 26 are the same.

In some references the authors use 'et al' (e.g. ref No 6 or 21), but in other references a long list of authors is included (e.g. ref No 30).

Philip et al. (2022) in the text but (2021) in Ref. list

Text has been carefully read through – we hope adequate corrections have been made. Reference list has been updated, with care taken to ensure consistent formatting in the journal style.